# A Smartphone-Based Pilot HIV Prevention Intervention (*Sakhi*) among Transgender Women Who Engage in Sex Work in India: Efficacy of a Pre- and Post-Test Quasi-Experimental Trial

**Venkatesan Chakrapani** [1,*] , **Pushpesh Kumar** [2] , **Jasvir Kaur** [3] , **Murali Shunmugam** [1] **and Debomita Mukherjee** [2]

1 Centre for Sexuality and Health Research and Policy (C-SHaRP), Chennai 600101, India
2 Department of Sociology, University of Hyderabad, Hyderabad 500406, India
3 Department of Nursing, Post-Graduate Institute of Medical Education and Research, Chandigarh 160012, India
* Correspondence: venkatesan.chakrapani@gmail.com

**Abstract:** Transgender women (TGW) in India, especially those who engage in sex work, are at high risk for HIV. Guided by the information-motivation-behavioral skills model and qualitative formative research findings, *Sakhi* (girlfriend), a 3-week smartphone-based pilot intervention consisting of short videos (one/week) and text messages (two/week), was implemented using a one-group pre- and post-test design to test its efficacy in promoting condom use and HIV testing among TGW (n = 50) who engage in sex work in Chennai. Changes in outcomes were assessed by conducting multivariable analyses using generalized estimating equations. Participants' mean age was 26 years, and the mean monthly income was INR 21700 (USD 292). About one-third completed college, and 96% were HIV-negative. Significant changes in the desired direction were observed in the primary outcomes: condom use – decrease in the engagement of condomless anal sex with male partners (12% to 2%, $p < 0.05$) and HIV testing – increase in intentions to undergo HIV testing every 6 months (34% to 86%, $p < 0.001$); and in some of the secondary outcomes: decrease in alcohol use before sex, increase in intentions to use condoms consistently and increase in the well-being score. This study demonstrated the feasibility, acceptability, and preliminary efficacy of the *Sakhi* intervention and warrants a larger randomized trial among diverse subgroups in diverse settings.

**Keywords:** condom use; HIV testing; mental health; e-health; m-health; outcome evaluation

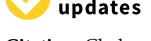



## 1. Introduction

Globally, transgender women (TGW) face a disproportionate HIV burden [1,2], with a much higher burden among those in sex work (19% vs. 27%) [3]. In India, a high proportion (60 to 70%) of TGW engage in survival sex work due to social exclusion and a lack of employment opportunities [4,5]. India's National AIDS Control Organisation (NACO) reports a high national average HIV prevalence among TGW ranging from 3.1% [6] to 9.5% [5], 15 to 40 times higher than that among the general population (0.2%); however, such estimates among those in sex work are not known. As postulated by the gender minority stress framework [7] and syndemic theory [1,8], the high levels of HIV prevalence among TGW in India could be because of high levels of societal stigma and marginalization faced by them, which contributes to psychosocial health problems such as violence victimization [9] and problematic alcohol use [10], which synergistically interact to increase HIV risk [10,11]. Social support and resilient coping (i.e., the tendency to cope adaptively with psychological stress) have been shown to mediate the effect of transgender identity stigma on depression [12]. Further, low social support has been shown to predict HIV risk among transgender women in India [10].

Consistent use of condoms and periodic HIV testing are essential elements of HIV prevention. As mentioned earlier, there is a lack of data on these elements for TGW who engage in sex work. However, despite more than a decade of government-supported

targeted HIV interventions among TGW, consistent condom use in anal sex with male paying partners remains low at 65% [13] – an indication of even higher levels of condomless anal sex among those in sex work. Additionally, there is a low uptake of HIV testing among TGW despite the fact that India has committed to achieving the goal of 95-95-95 (the first 95 being 95% of people with HIV know their HIV status) of UNAIDS/WHO by 2025 [14]. NACO recommends HIV testing every six months for at-risk populations, including TGW, while only 48.9% reported having ever had voluntary HIV testing as per a national survey [13]. There is a lack of information at the national level on the proportion of TGW, including those in sex work who test for HIV every six months. However, the available information shows that they face several individual and institutional level barriers for HIV testing due to transprejudice, anticipated stigma if tested as HIV positive, and discrimination in healthcare settings, including government HIV testing centers [15,16].

TGW in India are shifting to online spaces or using phones to seek sexual partners [17–19]. A national survey has also documented that 86% of TGW reported that their paying partners contacted them over mobile phones, and 25% used the Internet to contact their male clients [13]. Systematic reviews on the effectiveness of technology-based (mostly web-based or web + phone) interventions among vulnerable populations, including TGW [20,21] have shown mixed results. A few studies from Asia [22,23] have shown that online interventions delivered through mainstream social media-based platforms, mobile apps or online peer educators are useful to improve HIV prevention knowledge and HIV testing uptake and promote safer sex among sexual and gender minorities, including TGW. However, there is a lack of evidence on the efficacy of such interventions among TGW who engage in sex work. The 2017 strategic plan of India's National AIDS Control Organisation (NACO) has stated the importance of initiating online interventions for at-risk populations and to train peers on mobile and internet social and sexual networks [24]. Despite this, no online HIV-prevention interventions have been developed and tested among TGW who engage in sex work in India.

High levels of HIV prevalence, inconsistent condom use, and suboptimal HIV testing rates among TGW who engage in sex work, coupled with increasing use of mobile phone and the Internet to seek paying male sexual partners, strongly calls for evidence-based interventions that can be delivered online through smartphones. This study aims to address this gap. Accordingly, this study tested the efficacy of a smartphone-based HIV prevention intervention among TGW who engage in sex work to reduce condomless anal sex and to improve HIV testing and mental health.

## 2. Materials and Methods

### 2.1. Study Setting and Design

During June and July 2021, using one group pre- and post-test design [25,26], we conducted a pilot intervention among 50 TGW who engage in sex work in Chennai, the capital city of Tamil Nadu, a Southern State of India. Participants were recruited through two community-based organizations (CBOs); one of the CBOs provides HIV prevention services to TGW through physical outreach, and the other CBO provides social services and engages in advocacy activities. This study was conducted during the COVID-19 pandemic amidst lockdowns. Given the disproportionate impact of COVID-19 on the physical and mental health of sexual and gender minorities [27], including those in India [28], we decided not to include a control group and chose the one group pre- and post-test design.

### 2.2. Intervention Development

Intervention development was guided by the information-motivation-behavioral skills (IMB) model of HIV prevention behavior [29] and qualitative formative research. In brief, the IMB model states that information, motivation, and behavioral skills are the fundamental determinants of HIV preventive behavior. The model specifies that change in behavior largely occurs as a result of effects on behavioral skills that come from information about HIV risk behaviors and motivation to change those risk behaviors. The IMB model has

been widely used in HIV prevention research [30]. A few IMB model-based interventions consisting of single or multiple-session workshops have been used for improving HIV-prevention knowledge and modifying risk-taking behaviors among truck drivers in India [31] and adolescents in South Africa [32]. In order to ensure a culturally-appropriate intervention for Indian TGW who engage in sex work, the intervention was designed through community consultations and in close collaboration with CBOs serving TGW in Chennai. We conducted six focus groups (n = 30) among TGW, with 5 to 7 participants in each focus group and four key informant in-depth interviews with TGW community leaders. Each focus group lasted for about one to 1.5 h, and each key informant interview for about 30 min. Participants reported that smartphones are increasingly used by TGW in urban areas, especially those in sex work, for socializing with peers, seeking paying sexual partners, and entertainment. WhatsApp, Facebook, and certain dating apps or websites (e.g., Grindr, Tinder, Locanto) were commonly used. Even those TGW who cannot read or type sent and received voice or video messages in WhatsApp and other apps. Participants indicated preferences for short videos and text messages with a frequency of once or twice per week. Further, they emphasized that the information should not be limited to HIV but should include mental health, violence and discrimination, and gender transition (hormone therapy and gender-affirmative surgery). These inputs helped in finalizing the intervention components (videos and text messages) and delivery mode (WhatsApp), and frequency.

### 2.3. Intervention Description

Overall, our IMB-based intervention, nicknamed *Sakhi* (which means 'girlfriend'), addressed the specific information, motivation, and behavioral skills deficits that we identified during the formative research (Table 1). The information and motivational messages were provided in WhatsApp videos and messages, and behavioral skills (e.g., the correct way of using condoms) were shown in videos. The three-week intervention consisted of 3 videos (one per week) and 6 messages (two per week) on three topics – HIV testing, condom use, and mental health (Table 1). Each video was about 5 min and had TGW actors and narrators, and animations. The messages were delivered individually, and the direction of messages was one-way, i.e., from the research team to the participants. Prior to intervention implementation, the two peers were trained on delivering the intervention (sending messages and videos) and on research ethics.

### 2.4. Participants and Recruitment

Using purposive sampling, based on the guidelines for determining sample size for pilot intervention projects [33], 50 TGW engaged in sex work were enrolled in the study during June and July 2021. Study inclusion criteria were: age—18 years and above; self-identification as a transgender woman or thirunangai (or other indigenous or non-binary identities), engaged in sex work in the past three months; ability to read and understand local (Tamil) language; and have access to an Internet-enabled mobile phone.

Recruitment was mainly by word-of-mouth with the support of peer recruiters who had connections with two agencies that work with gender minorities in Chennai. Information about the study was also shared through the Chennai trans-specific WhatsApp groups.

**Table 1.** Use of Information-Motivation-Behavioral Skills (IMB) model constructs in designing videos and messages for the smartphone-based *Sakhi* HIV prevention intervention for transgender women.

| IMB Model Constructs | Deficits Identified in the Qualitative Formative Research | How These Deficits were Addressed in the Videos and Text Messages Sent in the *Sakhi* Intervention |
|---|---|---|
| Information | **Condom use**<br><br>• Consistent condom use was less common among regular male partners when compared to casual or paying partners<br>• Lack of information about HIV pre-exposure prophylaxis (PrEP)<br>• Wearing double condoms is seen as useful<br>• Alcohol use before or during sex led to not using condoms<br><br>**HIV testing**<br><br>• Challenges in undergoing HIV testing at least every 6 months (as per national guidelines)<br>• Lack of information about the need to undergo HIV testing (first time and repeat HIV testing every 6 months for transfeminine people) among some people<br><br>**Mental health**<br><br>• Low self-esteem and lack of self-acceptance<br>• Fear of double or triple discrimination (on the basis of gender identity, sex work status, and HIV status)<br>• Negative societal attitudes toward transgender people<br>• Lack of access to mental health services | **Condom use**<br><br>• Condoms need to be consistently used with all types of male partners<br>• Combining condoms with other forms of new prevention methods, such as PrEP, may offer additional protection<br>• The use of double condoms should be avoided<br>• Consumption of alcohol before sex increases the chances of condomless sex. Avoid alcohol use before sex.<br><br>**HIV testing**<br><br>• Need to undergo HIV testing at least every 6 months (as per national guidelines)<br>• People who tested positive for HIV should start taking antiretroviral treatment (ART) as soon as possible<br><br>**Mental health**<br><br>• Seek help from a mental health professional if you feel down<br>• Join support groups for trans people<br>• Take pride in trans identities<br>• Contact community agencies who refer them to mental health professionals |
| Motivation | • Want to protect self and partners from HIV and STIs<br>• Perceived challenges in going for free HIV testing in a government healthcare facility<br>• Lack of focus on maintaining mental health<br>• Having suicidal ideation | • Consistent condom use protects self and partners from HIV and STIs<br>• Community agencies can help in HIV testing and in navigating government HIV testing centers<br>• Maintaining optimal mental health is important |
| Behavioral skills | • Condom negotiation with male partners, both primary and casual, is difficult or not fruitful<br>• Going for HIV testing every 6 months is seen as burdensome or unnecessary | • Demonstration of the correct way of using condoms<br>• Tips on how to negotiate condom use with casual and primary partners<br>• Role models in the videos stress the importance of regular HIV testing (at least every 6 months) and consistent condom use to promote self-efficacy in undergoing HIV testing and using condoms consistently |

Given the visual and remote nature of videos and text messages in this intervention, the emphasis was on information and motivation constructs of the IMB model.

### 2.5. Measures

Surveys were administered face-to-face by trained research interviewers using a structured questionnaire at baseline and at the end of one month (about one week from the last video or text message). All participants (100%) completed the follow-up survey. Both surveys lasted approximately 30 min. We took COVID-19 precautionary measures: all participants and interviewers wore face masks, used hand sanitizers, and maintained physical distancing. The venues for the surveys (e.g., home, a private room in CBO) were based on the participant's preferences and convenience. Information was collected on baseline characteristics such as age, income, highest level of education completed, engagement in sex work and sexual identity, HIV testing and status, use of recreational

drugs and tobacco, use of the Internet in meeting sexual partners, enrollment in government-supported targeted HIV prevention interventions and study outcomes. Validated measures from prior studies with TGW were used wherever possible to measure the outcomes.

The study's outcome measures are summarized in Table 2, including information on the number of items and example items used to measure each variable, coding or scoring of responses, and the type of variables (binary or continuous). The primary outcome variables are: (1) Safer sex: engagement in condomless anal sex with male partners during the past month; (2) HIV testing intention: intention to undergo HIV testing every six months. Secondary outcome variables are: intention to use condoms consistently, condom use self-efficacy [34], alcohol use before last sex, correct knowledge of government-recommended HIV testing frequency, i.e., every 6 months (single item), and mental health—mental well-being [35,36], past two-week depression (PHQ-2) [37,38], and past two-week anxiety (GAD-2) [39,40]. Relevant predictors of mental health and HIV risk included past three-month transgender identity stigma experiences [41], social support [42], and resilient coping [43]. The scores on items measuring each of the continuous variables were summed to create composite scores for the respective variables. Standard cut-offs ($\geq 3$) were used to create a dichotomous measure for depression [44] and anxiety [39].

**Table 2.** Summary of the study's key outcome and predictor variables.

| Variables | Number of Items Used to Measure the Variable | Question or An Example Item of the Scales Used | Range of Responses and Scores | Type of Variables |
|---|---|---|---|---|
| **Primary Outcome Variables** | | | | |
| Condom use: Condomless anal sex with male partners (past one month) | 1 | How often have you used condoms when you had anal sex with male non-regular partners in the past one month? | Yes for 'Always' No for other options (most of the time, sometimes, rarely, never) | Binary |
| HIV testing intention: Intention to undergo HIV testing every 6 months | 1 | From now on I will make sure I always undergo HIV testing at least every six months | Yes/No | Binary |
| **Secondary Outcome Variables** | | | | |
| Intention to use condoms consistently | 1 | Which of the following best applies to you? | Yes for 'From now on I will make sure I always use a condom whenever I have sex' No for other options | Binary |
| Condom use self-efficacy | 6 | I would insist on using condoms during sex, even if my partner didn't want to | 1 for 'Not at all true of me' to 5 for 'Completely true of me' Score range: 6–30 | Continuous |
| Alcohol use before last anal sex | 1 | Last time you had anal sex with a male non-regular partner, did you or your partner drink alcohol before or during sex? | Yes/No | Binary |
| Correct knowledge of government-recommended HIV testing frequency (i.e., every 6 months) | 1 | What is the government-recommended frequency of HIV testing for vulnerable persons like transgender women (thirunangai)? | Yes for 'every six months' No for other responses | Binary |

**Table 2.** *Cont.*

| Variables | Number of Items Used to Measure the Variable | Question or An Example Item of the Scales Used | Range of Responses and Scores | Type of Variables |
|---|---|---|---|---|
| **Mental Health** | | | | |
| Mental well-being | 7 | I have been feeling optimistic about the future | 1 for 'None of the time' to 5 for 'All of the time' Score range: 7–35 | |
| Depression (past two weeks)—PHQ9 | 2 | Little interest or pleasure in doing things | 1 for 'Not at all' to 4 for 'Nearly every day' Score range: 2–8 For dichotomizing the measure, a cut-off score of >=3 depression was considered as indicative of depression | Binary |
| Anxiety (past two weeks)—GAD2 | 2 | Feeling nervous, anxious, or on edge | 1 for 'Not at all' to 4 for 'Nearly every day' Score range: 2–8 For dichotomizing the measure, a cut-off score of $\geq$3 was considered as indicative of anxiety | Binary |
| **Predictors of Mental Health and HIV Risk** | | | | |
| Transgender identity stigma (past three months) | 9 | How often have you heard that transgender people (thirunangai) are not normal? | 0 for 'Never' to 3 for 'Many times' Score range: 0–27 | Continuous |
| Social support | 3 | How many people are so close to you that you can count on them if you have great personal problems? | Item 1: 1 for 'None' to 4 for 'More than 5 people' Item 2: 1 for 'None' to 5 for 'A lot' Item 3: 1 for 'Very difficult' to 5 for 'Very easy' Score range: 3–14 | Continuous |
| Resilient coping | 4 | I look for creative ways to alter difficult situations | 1 for 'Disagree' to 3 'Agree' Score range: 4–12 | Continuous |

Process outcomes included satisfaction with the intervention and perceived usefulness of the intervention. Satisfaction with the intervention was measured with three questions that assessed satisfaction with videos and messages (responses ranging from 1 for very dissatisfied to 5 for very satisfied) and duration of videos (responses ranging from 1 for too lengthy to 5 for perfect). The perceived usefulness of the intervention was assessed by four questions that assessed the usefulness of the content of videos and messages and whether they will recommend the intervention to others (responses ranging from 1 for strongly disagree to 4 for strongly agree). These responses were dichotomized for analysis.

*2.6. Data Analysis*

Descriptive statistics were used to summarize the baseline sociodemographic characteristics and key outcome variables. For categorical outcome variables (e.g., intentions to consistently use condoms or to undergo HIV testing every 6 months), McNemar tests were used and for continuous outcome variables (e.g., depression and anxiety scores), paired *t*-tests were used to examine differences between baseline and post-intervention assessments. In order to obtain population-level (marginal) inference about how the average

rates of the outcomes may change in the TGW study population, multivariable analyses were conducted using generalized estimating equations (GEE) by including key relevant predictors of mental health and HIV risk (e.g., transgender identity stigma score, education, income, social support, resilient coping). GEE is a commonly used approach to analyze repeated measures data as it does not require distributional assumptions and produces robust estimates [45]. Proportions of participants who were satisfied with the intervention and found it useful were reported. All tests were conducted using Stata version 16, with statistical significance determined at $p < 0.05$ level.

## 3. Results

### 3.1. Participants' Characteristics

A total of 50 TGW were enrolled and retained in the study. Their mean age was 26 years (SD 4.9; range: 19–38 years), and their mean monthly income was INR 21700 (USD 292). About one-third completed a college degree (32%), 88% were single, and 96% reported that their gender identity was known to everyone (Table 3). Most participants (96%) tested negative on their most recent HIV test, 18% reported using recreational drugs in the past three months, 24% were current tobacco users, almost all (96%) reported using social media apps to find sexual partners, and 32% reported being enrolled in government-supported targeted HIV prevention interventions. Baseline means and standard deviations for relevant control variables were: $14 \pm 5.2$ for the transgender identity stigma score, $6.9 \pm 2.9$ for the social support score, and $11.3 \pm 1.1$ for the resilient coping score.

**Table 3.** Baseline characteristics of transgender women enrolled in *Sakhi*, a 3-Week pilot smartphone-based HIV prevention intervention (n = 50).

| Characteristics | n = 50 |
|---|---|
| **Mean age in years (SD)** | 26 (4.9) |
| **Mean monthly income (SD)** | INR 21700 (6935); USD 292 (93) |
| **Number of non-regular male partners** | |
| Mean (SD) | 18 (13.9) |
| Median (IQR) | 15 (7.0, 26.2) |
| **Highest level of completed education, n (%)** | |
| Primary (5th grade) | 2 (4.0) |
| Elementary (8th grade) | 9 (18.0) |
| High school (10th grade) | 13 (26.0) |
| Higher secondary (12th grade) | 5 (10.0) |
| Completed diploma course | 5 (10.0) |
| Completed college degree | 16 (32.0) |
| **Gender-related identity, n (%)** | |
| Thirunangai (equivalent to 'transgender woman') | 34 (68.0) |
| Woman | 16 (32.0) |
| **Gender identity known to everyone or most people, n (%)** | 48 (96.0) |
| **Current relationship, n (%)** | |
| Currently single, no main partner | 44 (88.0) |
| Has a boyfriend | 6 (12.0) |
| **HIV status negative, n (%)** | 48 (96.0) |
| **Recreational drug use (Marijuana) in past 3 months, n (%)** | 8 (16.0) |
| **Current tobacco use, n (%)** | 12 (24.0) |
| **Use dating/social media apps to find sex partners** | 48 (96.0) |

### 3.2. Effect of the Intervention on Study Outcomes

Bivariate analyses showed significant changes in most of the primary and secondary outcome variables in hypothesized directions at 4-week post-intervention compared to baseline (Table 4). A significant increase was observed in the intentions to use condoms consistently and undergo HIV testing every six months, the correct knowledge of the government-recommended HIV testing frequency, condom use self-efficacy, and well-being scores (Table 4).

**Table 4.** Outcome evaluation of *Sakhi*, a 3-week smartphone-delivered pilot online HIV prevention intervention among transgender women (n = 50).

| Outcomes | Baseline n (%) or Mean (SD) | Post-Intervention at 4 Weeks n (%) or Mean (SD) | Change %/Mean (95% CI) | $\chi^2$ or *t*-test [a] | Unadjusted Effect Size [b] (95% CI) | Adjusted Effect Size [c] (95% CI) |
|---|---|---|---|---|---|---|
| **Primary outcomes** | | | | | | |
| Condom use: Condomless anal sex with male partners (Yes) | 6 (12.0) | 1 (2.0) | −10 (−22, 2.0) | 3.6 | 0.2 (0.004, 1.4) | 0.08 (0.01, 0.8) * |
| HIV testing intention: Intention to undergo HIV testing every 6 months (Yes) | 17 (34.0) | 43 (86.0) | 52 (33, 71) | 21.1 | 9.7 (3, 50) *** | 9.4 (3.0, 29.4) *** |
| **Secondary outcomes** | | | | | | |
| Intention to use condoms consistently (Yes) | 11(22.0) | 41 (82.0) | 60 (40, 80) | 23.7 | 8.5 (3, 33) *** | 18.3 (5.2, 63.8) *** |
| Condom use self-efficacy score [d] | 26.5 (3.2) | 29.8 (1.0) | 3.3 (2.4, 4.2) | 7.08 | 1.4 (0.9, 1.8) *** | 3.3 (2.4, 4.3) *** |
| Alcohol use before last anal sex (Yes) | 15 (30.0) | 9 (18.0) | −12 (−28, 4) | 2.6 | 0.4 (0.1, 1.4) | 0.3 (0.1, 0.8) * |
| Correct knowledge of government-recommended HIV testing frequency (i.e., every 6 months) | 12 (24.0) | 28 (56.0) | 32 (10, 54) | 8.5 | 3.3 (1.4, 9.1) ** | 5.1 (1.8, 14.5) ** |
| Mental well-being score [d] | 23.9 (3.8) | 27.7 (2.1) | 3.8 (2.7, 4.8) | 7.3 | 1.2 (0.8, 1.6) *** | 4.2 (3.1, 5.3) *** |
| Depression (Yes) [e] | 13 (26.0) | 13 (26.0) | 0 | - | 1.0 (0.4, 2.5) | 0.9 (0.3, 2.6) |
| Anxiety (Yes) [f] | 18 (36.0) | 30 (60.0) | 24 (2, 46) | 5.1 | 2.5 (1.1, 6.6)* | 2.7 (0.9, 7.9) |

[a] McNemar's test for binary outcomes. Paired *t*-test for continuous outcomes. [b] Odds ratio for binary outcomes and Cohen's d for continuous outcomes. [c] In multivariable analyses, all outcome models were adjusted for transgender identity stigma score, education and income. For depression and anxiety outcome models, social support and resilient coping were also included as independent variables. [d] Higher score indicate better self-efficacy and well-being. [e] A cut-off of ≥3 was used as indicative of depression. [f] A cut-off of ≥3 was used as indicative of anxiety. * $p < 0.05$, ** $p < 0.01$, *** $p < 0.001$.

Multivariable analyses (Table 4) showed that post-intervention, the participants were significantly less likely to engage in condomless anal sex with male partners and more likely to have intentions to undergo HIV testing every six months (primary outcomes). Similarly, post-intervention, participants were less likely to use alcohol before sex, were more likely to have intentions to use condoms consistently, had higher scores on condom use self-efficacy, and were more likely to have correct knowledge of government-recommended HIV testing frequency (i.e., every 6 months) and better mental well-being compared to baseline. A one-unit increase in transgender identity stigma score was associated with a 14-point decrease in well-being (b coefficient: −0.14, 95% CI −0.26, −0.01; $p = 0.03$). The intervention had no effect on depression; however, each unit increase in social support was associated with a 30% lower likelihood of having depression (odds ratio: 0.7; 95% CI 0.5, 0.9; $p = 0.02$). Compared to bivariate analyses, adjusted analyses suggest a non-significant increase in the odds of anxiety at post-intervention compared to baseline (odds ratio: 2.7; 95% CI 0.9, 7.9; $p = 0.06$). However, a higher transgender identity stigma score was associated with a 20% higher likelihood of having anxiety (odds ratio: 1.20; 95% CI 1.02, 1.33; $p = 0.02$). Whereas higher social support and resilient coping scores were associated with a 30% and 50% lower likelihood of having anxiety, respectively.

### 3.3. Process Outcomes

Most of the participants (98%) were satisfied with the videos and text messages shared, and all reported that the duration of the videos was just correct or perfect. Most (98%) felt that the information content of the videos and text messages met their needs, that they learned something new from the intervention, and would recommend the intervention to others.

### 4. Discussion

The findings from this pilot smartphone-based intervention (*Sakhi*) demonstrated the feasibility of rapid recruitment and retention of a high-risk sample of TGW engaged in sex work. We found evidence for the preliminary efficacy of *Sakhi* intervention: a significant increase in intentions to undergo HIV testing every six months and to use con-

doms consistently; and a significant reduction in condomless anal sex with male partners. Although well-being scores significantly increased, this intervention seemed to have no effect on depressive and anxiety symptoms. Recruitment and follow-up by peers affiliated with local agencies working with trans people in Chennai and the relatively short period of intervention (one month) helped us in retaining all participants in the study without any attrition.

In India, a few online HIV prevention interventions have been conducted among men who have sex with men [46,47], but none among TGW. Given the lack of literature on effective culturally-appropriate online health interventions among TGW who engage in sex work in India, the evidence for the feasibility, acceptability, and preliminary efficacy of this pilot intervention is crucial. NACO reports that only 67% of at-risk TGW in India have been reached through its targeted HIV prevention interventions [48]. With nearly 85% of TGW, including those in sex work, reporting that their male partners contacted them over the phone and with 25% being reached through the Internet, smartphone-based online interventions offer a cost-effective way to reach the unreached TGW who engage in sex work [13]. The use of videos and text messaging to improve information and skills related to condom use, HIV testing, and mental health reduces the burden on human resources and the time burden of service providers and can complement the efforts of traditional venue-based HIV prevention outreach. However, lack of access to smartphones and limited digital literacy among several TGW who engage in sex work may pose challenges for scaling up smartphone-based online interventions. This situation may already be changing—increasingly intuitive and user-friendly features, drop in prices, and diverse uses, including meeting potential sexual partners virtually, make smartphones a necessity for TGW who engage in sex work. Further, even those with low literacy levels can send and receive voice and video messages through smartphones. Taken together, this means smartphone-based online interventions can be useful for even those TGW who are less literate.

The significant increase in the proportion of those who reported intentions to get tested for HIV every 6 months could possibly be through the significant increase in the correct knowledge of government-recommended HIV testing frequency, i.e., every six months. These findings are consistent with other studies among TGW that have shown an increase in HIV testing after video or social media interventions [20,49]. In relation to mental health outcomes, although there was a significant increase in well-being scores, there was no significant decrease in the proportion of those with depressive or anxiety symptoms. This could be because of both transgender identity stigma experiences as well as the mental health impact of COVID-19, which is connected to the loss of livelihood, especially for sex workers [28]. In fact, transgender identity stigma was positively associated with anxiety in the present study, possibly reflecting the impact of gender minority stress on psychological distress – consistent with the gender minority stress model [50]. Further, depression and anxiety may require relatively intensive intervention in the form of peer-delivered counseling or referrals to mental health counseling.

This intervention also resulted in a significant increase in consistent condom use with male partners and a significant increase in the proportion of those with intentions to use condoms consistently. It is possible that intervention led to a significant increase in condom use self-efficacy, which in turn led to a decrease in inconsistent condom use. Such an association has been found among men who have sex with men in India [51]. Given the small sample size, we could not conduct mediation analyses to test this hypothesis in this study.

*Strengths and Limitations*

The strengths of the study include the engagement of transfeminine communities in designing and implementing this theory-guided (IMB model) intervention, 100% retention rate, and high feasibility for scaling up the intervention. However, this study has several limitations. First, as a proof-of-concept pilot study, the results of this quasi-experimental study, including acceptability and feasibility, need to be tested in a larger trial that is

powered to conduct subgroup analyses. By the very nature of the study design, the selection bias in recruitment and lack of a control group points towards a possibility that the changes observed in the outcomes were unrelated to the intervention. However, all participants reported not receiving any other interventions, including those enrolled in government-supported HIV prevention interventions that involve venue-based outreach. Informal discussions with participants revealed that COVID-19 pandemic-related travel restrictions prevented them from interacting with the outreach staff of government-supported targeted HIV prevention interventions. Therefore, the reported changes could be attributed to this pilot intervention itself. Further, no change observed in variables that were not part of the intervention (e.g., sex work stigma score, social support score) provides further evidence for the same. Second, the generalisability of this pilot study is limited due to its inherent small sample size. The findings of this study may not be generalizable, especially to TGW who are part of traditional HIV prevention interventions, who are relatively less educated, and those living in rural areas. These diverse subgroups need to be included in the evaluation of a larger trial, preferably a randomized controlled trial. Third, because of resource and time constraints and due to the COVID-19 pandemic, the intervention videos and messages were not tailored to individuals, but the content of the videos and messages covered necessary information for participants in all stages of change described in the transtheoretical model of behavioral change [52]. Future interventions can tailor the intervention content according to the stage of change and incorporate other modes of delivery, such as tailored virtual peer-delivered counseling.

## 5. Conclusions

The pilot smartphone-based *Sakhi* intervention was found to be feasible and acceptable to TGW who engage in sex work and contributed to significant improvement in condom use, HIV testing intentions, and well-being. This study adds to the limited literature on the efficacy of smartphone-based online interventions among TGW who engage in sex work in India and supports the need for a randomized controlled study in diverse settings, with additional (e.g., virtual peer-delivered counseling) and tailored intervention components, and adequate power to detect subgroup differences. This study has also shown that smartphone-based online interventions are likely to especially useful in reaching TGW who engage in sex work who are not part of government-supported venue-based HIV prevention interventions and in offering health promotion messages beyond HIV.

**Author Contributions:** Conceptualization and Methodology, V.C. and P.K.; Data Curation, M.S. and D.M.; Software and formal analysis, V.C. and J.K.; Writing—Original Draft Preparation, V.C. and M.S.; Writing—Review & Editing, V.C., J.K. and M.S.; Supervision, P.K.; Project Administration, D.M.; Funding Acquisition, P.K. and V.C. All authors have read and agreed to the published version of the manuscript.

**Funding:** This study was funded by the Indian Council of Social Science Research (ICSSR) Impactful Policy Research in Social Science (IMPRESS) scheme (IMPRESS/P2830/591/2018-19).

**Institutional Review Board Statement:** The institutional review board of the Centre for Sexuality and Health Research and Policy (C-SHaRP) approved this study.

**Informed Consent Statement:** Written informed consent was obtained from all participants. Participants received INR 500 (~USD 6) each in pre- and post-intervention assessment surveys.

**Data Availability Statement:** The data that support the findings of this study are available from the second author upon reasonable request.

**Acknowledgments:** We gratefully acknowledge the support from Sahodaran and Thozhi, our partner agencies in Chennai, for the successful implementation and completion of this study. We thank all our study participants, without whom this study would not have been possible. We thank the community leaders—Jaya from Sahodaran and Sudha from Thozhi—for their inputs in the intervention design and for actively taking part in the development of videos and messages. We thank Visaka and Sundar for coordinating data collection activities in Chennai, and Padamavathi, Consultant, who directed

and filmed the videos. Chakrapani was supported by the DBT/Wellcome Trust India Alliance Senior Fellowship (IA/CPHS/16/1/502667).

**Conflicts of Interest:** The authors declare no conflict of interest.

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
