# Peer review of "A Smartphone-Based Pilot HIV Prevention Intervention (Sakhi) among Transgender Women Who Engage in Sex Work in India: Efficacy of a Pre- and Post-Test Quasi-Experimental Trial"

_venereology, doi:10.3390/venereology2010003_

Round 1
Reviewer 1 Report
This is an interesting manuscript that provides important information about a potentially scalable HIV prevention intervention for a high risk population. The authors provide a clear and concise background and justification for the intervention. Intervention development is clear. There are a few areas within the methods and procedures that would benefit from some clarification.
1) the authors state the aim - this study tested the efficacy of a smartphone-based HIV prevention intervention among TGW in sex work, to reduce condomless anal sex, and to improve HIV testing and mental health. Mental health is not clearly defined. Thee is mental well-being measure, a depression and anxiety measure - all could be mental health. Clearly operationalizing this important outcome variable seems essential to understand the procedures and results.
2) Is there a justification of sample size/power estimate for the 50 participants to detect adequate effect sizes?
3) The measure section was a bit confusing at times. The measure section stated "Relevant control variables included past three months transgender identity stigma (Chakrapani, Vijin, et al., 2017), social support (Kocalevent et al., 2018), and resilient coping (Sinclair & Wallston, 2004)" however the Data analysis stated "Whereas, higher social support and resilient coping scores were associated with a 30% and 50% lower likelihood of having anxiety, respectively" this indicated that these variables were not control variables, unless this reviewer missed something in the narrative.
4) Further, the justification for these two variables as control or predictor variables is not fully supported in the background nor the intervention development section.
5) please include information on how all the control variables were measured measured
Author Response
Thanks for your peer review work, please see our response at the attached file.

Reviewer 2 Report
Summary Statement: The authors conducted an HIV prevention intervention among transgender women in sex work. After the 3-week intervention, they found a decrease in condomless sex with male partners and an increased intention to undergo HIV testing every 6 months; they also found a decrease n alcohol use before sex and an increased intention to use condoms. As the few interventions among TGW who engage in sex work, this is vital work. A few comments to strengthen the position need to be addressed. Please see specific questions/comments below. I also added comments in the pdf attached.
Major comments:
1. Introduction: The author's a reference syndemic theory; however, there isn't any mention of Merrill Singer, who developed the theory in 1996. Please add Singer, M. (1996). A dose of drugs, a touch of violence, a case of AIDS: Conceptualizing the SAVA syndemic. Free Inquiry in Creative Sociology,24,99–110.
2. Introduction: Please update to 95-95-95 as we have already passed 2020 and now have new target goals
3. Introduction: Try to use language like TGW who engage in sex work rather than TGW in sex work
4. Method: There needs to be more specific detail about the TGW and key informant focus groups. How many focus groups? For how long were these conducted? How many participants per group?
5. Table 1: It might be helpful to have two different tables. One table with the qualitative formative research, and another one addresses elements of the Sakhi intervention. The table is difficult to understand as one table.
6. Methods: During the administration of the surveys, it would be helpful to have some details on what type of precautions were taken during COVID-19 for the interviewers and participants.
7. Discussion: Even though this is the first study among TGW who engage in sex work, it would be helpful to cite other mobile HIV prevention studies conducted in India among gender and sexual minorities.

Author Response

(The authors gave the same response as above.)

Round 2
Reviewer 1 Report
The Authors have done a nice job addressing the reviewers comments. I recommend publication.